A new large-bodied thalattosuchian crocodyliform from the Lower Jurassic (Toarcian) of Hungary, with further evidence of the mosaic acquisition of marine adaptations in Metriorhynchoidea

Ősi Attila 1 2 hungaros@gmail.com
http://orcid.org/0000-0002-7263-6505 Young Mark T. 3
Galácz András 1
Rabi Márton 4 5
1 Department of Paleontology, Eötvös University , Budapest , Hungary
2 Department of Paleontology and Geology, Hungarian Natural History Museum , Budapest , Hungary
3 Grant Institute, School of Geosciences, University of Edinburgh , Edinburgh , UK
4 Central Natural Science Collections, Martin-Luther Universität Halle-Wittenberg , Halle , Germany
5 Institute for Geosciences, University of Tübingen , Tübingen , Germany
Sues Hans-Dieter
Electronic publication date: 2018 May 10
Publication date: 2018
Volume: 6
Electronic Location ID: e4668
Received 2018 Feb 12; Accepted 2018 Apr 5
Copyright: © 2018 Ősi et al.
Copyright year: 2018
Copyright holder: Ősi et al.
License: This is an open access article distributed under the terms of the Creative Commons Attribution License, which permits unrestricted use, distribution, reproduction and adaptation in any medium and for any purpose provided that it is properly attributed. For attribution, the original author(s), title, publication source (PeerJ) and either DOI or URL of the article must be cited.
License URL: https://creativecommons.org/licenses/by/4.0/

Keywords: Crocodyliformes, Metriorhynchoidea, Marine adaptation, Hungary, Toarcian

Funding: National Research, Development and Innovation Office, Hungary OTKA K 116665 Eötvös University D11201/17 European Union’s Seventh Framework programme for research and innovation under the Marie Skłodowska-Curie 609402–2020 Volkswagen Foundation SYNTHESYS project FR-TAF-4021 DE-TAF-5132 DE-TAF-5698 HU-TAF-6505 European Community Research Infrastructure Action under the FP7 ‘Capacities’ programme Leverhulme Trust Research Project RPG-2017-167 Royal Society Research RG130018 This work was supported by the National Research, Development and Innovation Office, Hungary (OTKA K 116665) and Eötvös University (D11201/17). This project has received funding from the European Union’s Seventh Framework programme for research and innovation under the Marie Skłodowska-Curie grant agreement No. 609402–2020 researchers: Train to Move (T2M) to Márton Rabi. Márton Rabi is supported by a Volkswagen Foundation grant ‘Research in Museums.’ Mark T. Young received support for his collection visits to Paris (FR-TAF-4021), Stuttgart (DE-TAF-5132), Berlin (DE-TAF-5698) and Budapest (HU-TAF-6505) from the SYNTHESYS project http://www.synthesys.info/, which is financed by the European Community Research Infrastructure Action under the FP7 ‘Capacities’ programme. Mark T. Young is also supported by a Leverhulme Trust Research Project grant (RPG-2017-167), and the University of Edinburgh lab group is supported by a Royal Society Research Grant (RG130018). The funders had no role in study design, data collection and analysis, decision to publish, or preparation of the manuscript.

==============================
Based on associated and three-dimensionally preserved cranial and postcranial remains, a new thalattosuchian crocodyliform, Magyarosuchus fitosi gen. et sp. nov. from the Lower Jurassic (Upper Toarcian) Kisgerecse Marl Formation, Gerecse Mountains, Hungary is described here. Phylogenetic analyses using three different datasets indicate that M. fitosi is the sister taxon of Pelagosaurus typus forming together the basal-most sub-clade of Metriorhynchoidea. With an estimated body length of 4.67–4.83 m M. fitosi is the largest known non-metriorhynchid metriorhynchoid. Besides expanding Early Jurassic thalattosuchian diversity, the new specimen is of great importance since, unlike most contemporaneous estuarine, lagoonal or coastal thalattosuchians, it comes from an ‘ammonitico rosso’ type pelagic deposit of the Mediterranean region of the Tethys. A distal caudal vertebra having an unusually elongate and dorsally projected neural spine implies the presence of at least a rudimentary hypocercal tail fin and a slight ventral displacement of the distal caudal vertebral column in this basal metriorhynchoid. The combination of retaining heavy dorsal and ventral armors and having a slight hypocercal tail is unique, further highlighting the mosaic manner of marine adaptations in Metriorhynchoidea.

Introduction

The Early Jurassic was a critical period in the initial development of marine adaptation in crocodylomorphs (Wilberg, 2015a). Whereas the small-bodied, cursorial protosuchians existed on land (Colbert & Mook, 1951) and the nearshore to fluvial environments were inhabited by semi-aquatic goniopholidids (Tykoski et al., 2002), the first thalattosuchians appeared with the basal-most forms already showing a high number of anatomical traits suitable for a predominantly marine lifestyle (Young et al., 2010; Wilberg, 2015a; Bronzati, Montefeltro & Langer, 2015). Thalattosuchians are composed of two major groups, the teleosauroids and metriorhynchoids (Buffetaut, 1980; Young & Andrade, 2009; Young et al., 2010). Although teleosauroids were not as well-adapted to marine habitats as metriorhynchoids, their reduction in limb size and osteoderms (Buffetaut, 1980, 1982; Young et al., 2016) coupled with a gracile and streamlined body, that had a relatively rigid skeleton capable of sub-undulatory swimming (Massare, 1988; Hua & Buffetaut, 1997), clearly shows that they were efficient swimmers. Metriorhynchoids, and especially metriorhynchids, on the other hand, became even more adapted to a marine lifestyle, evolving paddle-like limbs, hypocercal tail fin, enlarged preorbital salt glands, and osteoporotic-like bone tissues (Fraas, 1902; Andrews, 1913; Hua & Buffrénil, 1996; Fernández & Gasparini, 2008; Young et al., 2010; Wilberg, 2015a).

In the summer of 1996, a partial skeleton of a thalattosuchian crocodyliform from the Lower Jurassic Kisgerecse Marl Formation of northwestern Hungary was discovered (Kordos, 1998; Ősi et al., 2010). We present a detailed osteological work and a series of extensive phylogenetic analyses of this fossil and assign it to a new genus and species. Besides expanding Early Jurassic thalattosuchian diversity, the new specimen is of great interest since, unlike most contemporaneous estuarine, lagoonal or coastal thalattosuchians it comes from an ‘ammonitico rosso’ type pelagic deposit.

Geological Setting and Paleoenvironment

The specimen was collected in one of the northwestern quarries of the Nagy-Pisznice Hill, close to Békás-Canyon (GPS coordinates: 47°42′09.4″N, 18°29′40.0″E), eastern Gerecse Mountains, northwestern Hungary (Fig. 1).

Figure 1 Locality map of the new thalattosuchian crocodyliform, Magyarosuchus fitosi gen. et sp. nov. from the Toarcian of the Gerecse Mountains, Hungary.

Red point marks the fossil site.

The remains of this large-bodied crocodyliform came from a fossiliferous limestone with a well-constrained stratigraphy (Galácz et al., 2010). These beds also yielded diagnostic ammonites, including Grammoceras thouarsense (d’Orbigny, 1842–51) which is an index fossil of the Upper Toarcian (Lower Jurassic) G. thouarsense ammonite Zone. In lithostratigraphic terms, the bed yielding the vertebrate remains (Bed 13) corresponds to the uppermost section of the Kisgerecse Marl Formation (Fig. 2), a red, nodular clayey limestone widely distributed in the Gerecse Mountains (Császár, Galácz & Vörös, 1998). The overlaying beds belong to the Tölgyhát Limestone Formation, representing the uppermost Toarcian and the Aalenian–Bajocian in the Eastern Gerecse (Cresta & Galácz, 1990). The Kisgerecse Marl and the Tölgyhát Limestone Formations are members of the Jurassic calcareous sequence that is interrupted only by a few meters of siliceous radiolarite in the Middle Jurassic (Fodor & Főzy, 2013a; Fig. 2). The locality is in the eastern part of the Gerecse Mountains, which was a deeper, basinal area east to the Jurassic–Early Cretaceous submarine high (the ‘Gorba High’) in the western part of the mountains (see Vörös & Galácz, 1998).

Figure 2 Schematic geological section of the locality at the Nagy-Pisznice Hill, close to Békás-Canyon (GPS coordinates: 47°42′09.4″N, 18°29′40.0″E), eastern Gerecse Mountains, northwestern Hungary.

The Upper Toarcian fossiliferous bed (Bed 13) produced the remains of the new thalattosuchian Magyarosuchus fitosi gen. et sp. nov.

The Jurassic of the Gerecse Mountains belongs to the Transdanubian Range of the Alpaca unit within the Alp-Carpathian framework (Fodor & Főzy, 2013b). The whole Jurassic sequence of the Gerecse is built up by pelagic carbonates which form a succession of reduced thickness and incomplete stratigraphic representation. This means that some stratigraphic units of subzonal or zonal rank may be missing in sections and these hiati are indicated by so-called hard grounds, suggesting interruptions in sedimentation. All these phenomena are characteristic in these carbonate sequences of the Mediterranean region of the Mesozoic Tethys, where the dominant rocks are the so-called rosso ammonitico limestones and marls. These sequences are interpreted as deposited in the pelagic realm, on deeply submerged continental slope, far away from continental land masses, thus free of clastic material influx (see Bernoulli & Jenkyns, 2009). Pelagic environment with a comparatively deep-water depth is indicated also by the faunal composition of the ammonitico rosso type rocks: elements of benthic invertebrates are represented in insignificant amount (sporadic bivalves and brachiopods), and the single frequent group is of the nectonic cephalopods. The cephalopods, dominated by ammonoids, appear in associations where the major groups are the phylloceratids and lytoceratids. These ammonite faunal compositions clearly indicate open marine environments with oceanic water depths with at least a few hundred meters (Westermann, 1990; Lukeneder, 2015).

Material, Preservation and Methods

Material. The vertebrate material consists of a partial skeleton of a large-sized thalattosuchian crocodyliform including both cranial and postcranial remains. All the specimens are housed in the Vertebrate Collection of the Department of Paleontology and Geology of the Hungarian Natural History Museum (MTM). Unfortunately, detailed information on the circumstances of the fieldwork is not available. A very rough sketch of the specimen has been drawn during the work, but is not applicable for taking precise measurements.

Preservation. Since many Early Jurassic thalattosuchians (such as those of Steneosaurus bollensis and Pelagosaurus typus; e.g., Westphal, 1962) are known from flattened specimens preserved in laminated shale (Posidonia Shale), the three-dimensional preservation makes the new specimen particularly important. Furthermore, in many cases the finest details of skeletal anatomy, such as the shallow crest-like edges of the attachment surface of the cartilage on the epiphyses have been also preserved by the hard limestone matrix. On the other hand, due to the very slow sedimentation rate of these highly condensed Lower Jurassic rocks (Bernoulli & Jenkyns, 2009), some of the bone surfaces were partially dissolved, as seen, for example on the femoral mid-shafts. Dissolution of fossils from these strata, however, is not rare: ammonite shells are frequently found to have a complete lower side and a partially or completely dissolved upper side.

Methods. Specimens have been prepared both mechanically and chemically. A Vibro-tool has been used for clearing the bones from the larger pieces of matrix. In some cases, chemical preparation using acetic acid was applied for a better cleaning of the bone surfaces.

The electronic version of this article in portable document format (PDF) will represent a published work according to the International Commission on Zoological Nomenclature (ICZN), and hence the new names contained in the electronic version are effectively published under that Code from the electronic edition alone. This published work and the nomenclatural acts it contains have been registered in ZooBank, the online registration system for the ICZN. The ZooBank LSIDs (Life Science Identifiers) can be resolved and the associated information viewed through any standard web browser by appending the LSID to the prefix http://zoobank.org/. The LSID for this publication is: [urn:lsid:zoobank.org:pub:3623D096-C737-4B69-A491-ABC0F50FF4D4]. The online version of this work is archived and available from the following digital repositories: PeerJ, PubMed Central and CLOCKSS.

Systematic Paleontology

CROCODYLOMORPHA Hay, 1930 (sensu Nesbitt, 2011)

THALATTOSUCHIA Fraas, 1901 (sensu Young & Andrade, 2009)

METRIORHYNCHOIDEA Fitzinger, 1843 (sensu Young & Andrade, 2009)

MAGYAROSUCHUS gen. nov.

Type species—Magyarosuchus fitosi gen. et sp. nov. (type by monotypy).

Etymology—‘Hungarian crocodile.’ Magyaro referring to the Hungarian people, and suchus is the Latinized form of the Greek soukhos (σoῦχoς), meaning crocodile.

Diagnosis—Same as the only known species (monotypic genus).

MAGYAROSUCHUS FITOSI, gen. et sp. nov.

Holotype—middle third of left dentary (V.97.2.A), posterior third of left dentary (V.97.2.B), mandible fragment (V.97.2.C), angular-dentary + surangular fragment (V.97.40.); 21 teeth (V.97.1., V.97.4., V.97.53., V.97.5., V.97.24., V.97.37., V.97.29., V.97.55., V.97.56.); three dorsal vertebrae (V.97.26., V.97.30.); two sacral vertebrae (V.97.30.); two proximal caudal vertebrae (V.97.29., V.97.30.); six mid-caudal vertebrae (V.97.27., V.97.28.); 12 distal caudal vertebrae (V.97.19., V.97.21., V.97.22., V.97.27., V.97.31.); 28 dorsal rib fragments (V.97.16., V.97.14., V.97.46., V.97.15., V.97.8., V.97.17., V.97.47., V.97.67., V.97.51., V.97.52., V.97.54., V.97.64., V.97.68., V.97.48., V.97.38.); sacral ribs (V.97.37., V.97.27.); coracoideum (V.97.7.); radius (V.97.42.); right ilium (V.97.44.); left ilium (V.97.34.); left ischium (V.97.36.); left pubis (V.97.49.); right pubis (V.97.35.); left femur (V.97.13.), right femur (V.97.33.); right tibia (V.97.9.); left tibia (V.97.69.); fibulae? (V.97.41., V.97.43.); four metapodial elements (V.97.10., V.97.11., V.97.38., V.97.45.); phalanges (V.97.61.); other limb bones (V.97.15.); four dorsal osteoderms (V.97.59., V.97.60.); 12 ventral osteoderms (V.97.18., V.97.38., V.97.65.); 27 fragmentary osteoderms (V.97.4., V.97.53., V.97.24., V.97.60., V.97.56.); other fragmentary elements: (V.97.49., V.97.50., V.97.58., V.97.60.). Note that in some cases, the same catalogue number belongs to different bones or teeth because blocks of rock contain more than one element and these blocks have been assigned to catalogue numbers. Measurements of the bones are listed in Table 1.

Table 1 Measurements of the bones of Magyarosuchus fitosi gen. et sp. nov.

Specimen no.	Skeletal element	Greatest diameter (mm)	
V.97.2A	Dentary fragment	143	
V.97.2B	Left dentary posterior fragment	171	
V.97.2C	Mandible fragment	128	
V.97.40	Left angular-surangular	106	
V.97.26	Dorsal vertebra	68	
V.97.26	Dorsal vertebra	67	
V.97.30	Last dorsal vertebra	58	
V.97.30	First sacral vertebra	60	
V.97.30	Second sacral vertebra	58	
V.97.30	First caudal vertebra	53	
V.97.29	Proximal caudal vertebra	60	
V.97.27	Mid-caudal vertebra	61	
V.97.28	Mid-caudal vertebra	63	
V.97.28	Mid-caudal vertebra	63	
V.97.28	Mid-caudal vertebra	61	
V.97.28	Mid-caudal vertebra	62	
V.97.27	Fragmentery distal caudal vertebra	58	
V.97.27	Distal caudal vertebra	61	
V.97.27	Distal caudal vertebra	63	
V.97.27	Distal caudal vertebra	62	
V.97.21	Distal caudal vertebra	62	
V.97.21	Distal caudal vertebra	64	
V.97.22	Distal caudal vertebra	63	
V.97.31	Distal caudal vertebra	60	
V.97.31	Distal caudal vertebra	59	
V.97.31	Distal caudal vertebra	63	
V.97.31	Distal caudal vertebra	59	
V.97.19	Last caudal vertebra	23	
V.97.37	Sacral rib with crest	74	
V.97.39	Sacral rib	75	
V.97.7	Right coracoid, fragment with coracoid foramen	66	
V.97.7	Right coracoid, distal half	80	
V.97.34	Left ilium	117	
V.97.35	Right pubis	164	
V.97.49	Distal half of left pubis	103	
V.97.36	Left ischium	137	
V.97.44	Right ilium	97	
V.97.33	Right femur	360	
V.97.13	Left femur	355	
V.97.69	Left tibia	213	
V.97.9	Right tibia	210	
V.97.15	Proximal fibula	62	
V.97.45	Metatarsal	61	
V.97.10	Metatarsal III	127	
V.97.11	Metatarsal	72	
V.97.12	Tarsus	36	
V.97.38	Ventral osteoderm	77	
V.97.59	Dorsal osteoderm	92	
V.97.60	Dorsal osteoderm	89	

Etymology—‘Fitos’s Hungarian crocodile.’ The name refers to Attila Fitos, discoverer of the specimen in thanks for his donation of the fossil to science.

Type locality—one of the northwestern quarries of the Nagy-Pisznice Hill, close to Békás-canyon (GPS coordinates: 47°42′09.4″N, 18°29′40.0″E), eastern Gerecse Mountains, northwestern Hungary.

Type horizon—Bed 13, Kisgerecse Marl Formation, Transdanubian Central Range. Grammoceras striatulum ammonite Subzone, G. thouarsense ammonite Zone, Upper Toarcian, Lower Jurassic (Galácz et al., 2010).

Diagnosis—Large-sized (estimated body length: in the range of 4.67–4.83 m) metriorhynchoid thalattosuchian with the following unique combination of characters (proposed autapomorphic characters are indicated by an asterisk (*)): tooth crown carinae development variable, being well-developed apically, beginning to develop mid-crown and absent in the basal region; enamel ornamentation is composed of ridges that differ in arrangement on the labial and lingual surfaces, being more widely spaced on the labial surface than the lingual surface, with the lingual surface having tightly packed apicobasal ridges basally which apically become shorter and discontinuous, and the apical lingual ridges on the mesial and distal margins bend towards the carinae (but do not contact them)*; abrupt change in centrum shape of the distal caudal vertebrae, with strong mediolateral compression (i.e. distal vertebrae are clearly heteromorphic); dorsal osteoderms have irregularly shaped pits (including circular, ellipsoid, bean-shaped, triangular and quadrangular shapes), with an extreme variation in size (from small to very large), with elongate pits present on the ventrolateral surface running from the keel to the lateral margin*; dorsal osteoderms have an anterolateral process that is ‘indistinct,’ no longer being distinctly ‘peg-like,’ as their lateral margin is contiguous with that of the osteoderm ventrolateral surface*.

Characteristics shared with Pelagosaurus. M. fitosi shares the following two synapomorphies with Pelagosaurus: (1) the surangulodentary and angulodentary grooves are parallel and positioned close to one another ventral to the dentary tooth row; and (2) the presence of a distinct anterior acetabular flange on the ilium, created by the anterior acetabular margin projecting anteriorly such that it is anterior to the iliac anterior margin. However, these two characters are currently unknown in all other basal metriorhynchoids and their distribution is therefore unknown. However, they are absent in teleosauroids and metriorhynchids (Fraas, 1902; Andrews, 1913; Johnson et al., 2017).

Metriorhynchoid characteristics shared. M. fitosi has the following two metriorhynchoid synapomorphies: (1) a coracoid with both the proximal and distal ends convex, and (2) the femur posteromedial tuber is present and the largest of the proximal tubera.

Description and Comparisons

Cranial elements

Mandible. Three fragments (MTM V.97.2.) of the left ramus of the mandible are preserved. Based on the mandibular proportions of P. typus (BRLSI M1413, Pierce & Benton, 2006) the first fragment (MTM V.97.2.A, Figs. 3A and 3B) most probably represents the middle third of the dentulous part of the dentary. There is no indication of any post-dentary bones preserved on this element. It has a dorsoventrally high profile. The ventral side is eroded and the medial side is covered with hard matrix, thus it is not clear whether this part formed already the symphyseal region. Laterally, it possesses an upper surangulodentary and a lower angulodentary groove parallel with each other, a feature that is also present in P. typus (BRLSI M1413, MTM M 62 2516). On the dorsal or dorsolateral side of the bone, six large (diameter: 10 mm) alveoli can be observed. They have an oblique, anterolabial–posterolingual orientation suggesting that the teeth oriented anterolabially or slightly dorsolabially instead of pointing simply dorsally. Interalveolar septa are anteroposteriorly thick (ca. 7–10 mm) reflecting widely spaced teeth in this part of the tooth row. Some pits as part of the lateral ornamentation can be observed, but it is not clear how developed this ornamentation was on the lateral side since this surface has been slightly dissolved due to diagenetic processes.

Figure 3 Mandibular elements of Magyarosuchus fitosi gen. et sp. nov. from the Toarcian of the Gerecse Mountains, Hungary.

(A), left dentary fragment (MTM V.97.2.A), middle portion in lateral; (B), dorsal views. (C), left dentary fragment (MTM V.97.2.B), posterior portion in lateral; (D), dorsal; (E), medial views. (F), dorsal (dentary + surangular) and ventral (angular) margins of the mandibular fenestra of the left mandible (MTM V.97.40) in dorsal; (G), medial; (H), lateral; (I), ventral views. Note that the specimen is dorsoventrally compressed, artificially closing the external mandibular fenestra. (J), Right? Mandible fragment (MTM V.97.2.C) in dorsal; (K), ventral; (L), lateral views. al, alveolus; an, angular; emf, external mandibular fenestra; gr, surangulodentary and angulodentary grooves; maf, mandibular adductor fossa; s, suture, sa, surangular; sh, shelf.

The second fragment (MTM V.97.2.B, Figs. 3C–3E) is the posterodorsal segment of the left dentary. Although there is no direct connection preserved with the dentary fragment described above, this second piece is apparently the posterior continuation of that element. Anterodorsally, it bears the last three alveoli which show a similar orientation and widely spaced configuration as those seen on the first fragment. Posterior to the last alveolus, the dentary becomes slightly elevated. Medially, the anterodorsal part of the deeply concave mandibular adductor fossa can be observed. The ventral side of the specimen is missing (Fig. 3E). The lateral surface is generally smooth, but on the ventral part some pits as part of the sculpture are present. In P. typus (BRLSI M1413) this part of the mandible was formed by the surangular and the dentary (Pierce & Benton, 2006; MTM M 62 2516). However, in M. fitosi the dentary–surangular suture cannot be detected. On the dorsal side, posterior to the last alveolus a shelf is present that has a slightly elevating medial side.

The third block (MTM V.97.40., Figs. 3F–3I) preserved from the left ramus of the mandible represents the dorsal and ventral margins of the external mandibular fenestra. Post-mortem deformation of the two bones resulted in the closure of the external mandibular fenestra, but the ventral margin of the surangular and the dorsal margin of the angular is smooth and not like a scarf joint indicating the presence of a most probably narrow, anteroposteriorly elongate opening between the two bones. The dorsal part is the middle portion of the surangular and the posterior process of the dentary and the ventral piece is the middle portion of the angular. On the dorsal piece, the dentary–surangular suture is observable. The lateral surface of the bones is smooth being completely avoid of the pitted ornamentation. Medially, they are concave forming the dorsal, lateral and ventral margins of the mandibular adductor fossa. Ventrally, the angular is widened forming a massive ventral bar of the postdentary part of the mandible. On its posteroventral surface, some grooves can be observed which, according to Iordansky (1973) and Mueller-Töwe (2006), should have served as the insertion of Musculus pterygoideus posterior.

A fourth element (MTM V.97.2.C; Figs. 3J–3L) is probably from the right ramus of the post-dentary part of the mandible. It is too fragmentary to tell more details on its position. Whereas its medial side partly preserves its original smooth surface, the outer surface is strongly eroded, only a small part shows some unornamented texture.

Teeth. Twenty-one teeth or tooth fragments have been preserved associated with the skeleton. These have conical and generally massive crowns (Fig. 4) being much more robust than the teeth of Pelagosaurus. They have a circular or sub-circular cross-section and some teeth are quite elongated with a crown height/width ratio over three (Figs. 4A and 4B), whereas others are stockier with a ratio of two or less (Figs. 4C–4F). In contrast to P. typus (MTM M 62 2516) but similar to Zoneait nargorum (Wilberg, 2015a), all teeth bear mesial and distal unserrated carinae which disappear towards the base of the crown (Figs. 4E–4I). The crown surface is ornamented by longitudinal enamel wrinkles on all sides (Figs. 4H and 4I) that are more prominent than those in S. bollensis (MTM M 69 242). Morphology of the posterior teeth are generally similar to those of Lemmysuchus obtusidens (Johnson et al., 2017), but they are devoid of any type of serration or anastomosing apical enamel wrinkles. Roots are preserved in most of the teeth and are two to three times longer than the crowns, and together with the crowns they are strongly curved lingually (Figs. 4C–4F). Wear pattern due to tooth–tooth contact cannot be observed on the tooth crowns.

Figure 4 Teeth of Magyarosuchus fitosi gen. et sp. nov. from the Toarcian of the Gerecse Mountains, Hungary.

(A), anterior tooth (MTM V.97.57.) with root fragment in labial view. (B), anterior or middle tooth (MTM V.97.57.) with root in mesial/distal view. (C), posterior tooth (MTM V.97.1) with root in labial; (D), lingual; (E), ?mesial; (F), distal views. (G), middle or posterior tooth (MTM V.97.57.) in mesial/distal views. (H and I), details of the ornamentation and the unserrated carina of (MTM V.97.57.), scale represents millimeter. c, carina; ec, end of the carina; wr, wrinkle.

Post-cranial axial skeleton

The vertebral column is not complete, represented only by three dorsal, two sacral, and 20 caudal vertebrae. All the vertebrae are platy- or slightly amphycoelous, and are devoid of pneumatic foramina. Neural arches are fully fused to the centra in all elements suggesting an ontogenetically mature individual. Cervical vertebrae seem to be not preserved in the material.

Dorsal vertebrae. The centrum of dorsal vertebrae (MTM V.97.26., MTM V.97.30) is higher than wide, moderately concave laterally and ventrally (Figs. 5A–5C). Its ventral surface is devoid of any grooves or crests. Anterior and posterior articulation surfaces are oval to slightly trapezoid in shape. Transverse processes emerge from the lateral side of the neural arch. The neural spine is rectangular in lateral view and its height is approximately three-fourth of that of the centrum. Anteroposteriorly, however, neural spines are relatively slightly shorter on the best preserved specimen (Fig. 5A) than that of Steneosaurus or Pelagosaurus.

Figure 5 Axial elements of Magyarosuchus fitosi gen. et sp. nov. from the Toarcian of the Gerecse Mountains, Hungary.

(A), dorsal vertebra (MTM V.97.26.) in right lateral; (B), ventral; (C), anterior views. (D), sacrum with the last dorsal and the first caudal vertebra (MTM V.97.30.) in right lateral; (E), ventral views. (F), middle caudal vertebra (MTM V.97.28.) in right lateral; (G), anterior; (H), ventral views. (I), distal caudal vertebra (MTM V.97.31.) in anterior, (J), right lateral views. (K), distal caudal vertebra (MTM V.97.19.) with massive neural spine in right lateral; (L), left lateral; (M), posterior; (N), anterior; (O), dorsal; (P), ventral views. ansp, anterior process of the neural spine; ca1, first caudal vertebra; ld, last dorsal vertebra; nc, neural canal; nsp, neural spine; pnsp, posterior process of neural spine; prz, prezygapophysis; sa1-2, sacral vertebrae 1-2; sra1-2, articulation for sacral ribs 1-2; trp, transverse process.

Sacral vertebrae. Sacral vertebrae (MTM V.97.30.) are preserved in a complex with the last dorsal and the first caudal vertebrae (Figs. 5D–5E). There are two true sacral vertebrae, as is the norm for thalattosuchians (Fraas, 1902; Andrews, 1913; Westphal, 1962), and in contrast to machimosaurin teleosauroids which have three due to the sacralisation of the first caudal vertebra (Andrews, 1913; Hua, 1999; Young et al., 2014; Johnson et al., 2017). Since a thin layer of sediment can be observed between the vertebrae, they were presumably not co-ossified. The two sacral vertebrae are quite similar to each other in having lateromedially wide and ventrally concave centra. Sacral ribs are not fused to the centra in contrast to the condition seen in S. bollensis (MTM M 69 242). Their articulation surface on the centra are large, anteroposteriorly elongated, oval shaped surfaces among which those of the anterior sacral are in an anterior and those of the posterior are in a posterior position. Whereas the basal, zygopophyseal articulations of the neural arches seem to be fused, the neural spines are separated, short processes.

Caudal vertebrae. The first caudal, being fused to the second sacral (MTM V.97.30., Figs. 5D and 5E), is longer than wide and as wide as high being ventrally very slightly concave, and the broken transverse processes are in an anterior position on the side of the centrum. The neural spine is shallow, having ca. half of the height of the centrum. The more posterior caudal vertebrae are more elongate and lateromedially slightly compressed with moderately concave lateral and ventral sides (Figs. 5F and 5G). Articular surfaces are oval in the mid-series caudal vertebrae, whereas they are rounded or slightly rectangular in the distal caudals. Transverse processes are positioned close to the centrum-neural arch fusion and posteriorly they become gradually shorter, laterally projecting processes. Similar to the 11th to the 22nd caudal vertebrae of P. typus (MTM M 62 2516), the neural spines of two of the preserved caudals of M. fitosi (MTM V.97.31.) are divided into a smaller, triangular, dorsally or anterodorsally projecting process and a larger, posterodorsally oriented process (Fig. 5J).

The distal-most preserved caudal (MTM V.97.19., Figs. 5K–5P) is the posterior half or two-third of the complete vertebra. Here, the neural spine is an anteroposteriorly wide, relatively massive, plate-like element. Its preserved part is as high as the centrum but is broken both anteriorly (Fig. 5N) and dorsally (Fig. 5O) indicating that it was originally much higher. In P. typus (MTM M 62 2516) and S. bollensis (MTM M 69 242), the distal caudals have only a posterodorsally projecting, anteroposteriorly narrow spine that emerges only on the posterior half of the centrum. The posterior articular surface of the centrum is close to quadrangular in shape and slightly concave.

Ribs. Numerous fragmentary ribs and rib fragments are preserved and they are generally similar to those of extant crocodylians. Two of them are interpreted as dorsal ribs in having an elongate, anteroposteriorly wide capitulum and a relatively short tuberculum (Figs. 6A and 6B). In cross-section they are close to oval shaped, but they bear a shallow crest on the posterior side of their proximal half that disappears distally. The dorsal ribs of Magyarosuchus are very similar to those of Steneosaurus (Andrews, 1913).

Figure 6 Appendicular elements of Magyarosuchus fitosi gen. et sp. nov. from the Toarcian of the Gerecse Mountains, Hungary.

(A), fragmentary dorsal rib (MTM V.97.8.) in anterior; (B), posterior views. (C), sacral rib (MTM V.97.37.) in anterior/posterior; (D), medial views. (E), sacral rib (MTM V.97.39.) in anterior/posterior; (F), medial views. (G), coracoid (MTM V.97.7.) in ventral view. (H), glenoid of the coracoid (MTM V.97.7.). (I), left ilium (MTM V.97.34.) in anterior; (J), laterodorsal; (K), posterior; (L), lateral views. (M), right ilium (MTM V.97.44.) in medial view. (N), left pubis (MTM V.97.35.) in anterodorsal; (O), lateral views. (P), distal half of the left ischium (MTM V.97.36.) in lateral view. ac, acetabulum; as, articulation surface; ca, capitulum; cf, coracoid foramen; gl, glenoid; ic, iliac crest; isa, articulation surface for ischium; prp, preacetabular process; pop, postacetabular process; sra, articulation surface for sacral rib; t, tuberculum.

Three of the sacral ribs are preserved (Figs. 6C–6F). All of them are short and massive with oval shaped (in MTM V.97.37. slightly rugose) articular surface. In anteroposterior view, they are triangular in shape with a slightly lateroventrally bent distal end. MTM V.97.37. is the largest, has a convex crest-like dorsal margin and probably represents the first sacral rib. The third specimen (MTM V.97.27.) is strongly eroded but the vertebral articulation is partly preserved. Sacral ribs of this Magyarosuchus differ from the elongate, slender sacral ribs of metriorhynchids, but are more similar to those of Pelagosaurus.

Appendicular skeleton

Coracoid. From the pectoral girdle elements only the right coracoid (MTM V.97.7.) is preserved in two pieces (Figs. 6G and 6H). It has the same bow-tie morphology as that of P. typus (Pierce & Benton, 2006) and S. bollensis (Westphal, 1962) with concave anterior and posterior margins and a convex ventral articulation surface. The medial surface is concave and partly resorbed due to diagenetic processes, whereas the lateral surface is smooth with a marked, oval-shaped coracoid foramen piercing it. The glenoid is a slightly convex surface. The distal half is strongly flattened and divergent ending dorsally in a convex edge.

Radius. From the forelimbs, only the proximal end of one radius (MTM V.97.42.) is preserved (Figs. 7A and 7B). The radius slightly widens towards the articular region and has a concave articular surface being similar to that of S. bollensis (MTM M 69 242) and Platysuchus multiscrobiculatus (SMNS 9930, Westphal, 1962).

Figure 7 Limb elements of Magyarosuchus fitosi gen. et sp. nov. from the Toarcian of the Gerecse Mountains, Hungary.

(A), proximal end of radius (MTM V.97.42.) in ?lateral; (B), ?medial views. (C), a short limb bone (?metacarpal or ?ulnare) with distal articular surface (left) associated with a dorsal rib (central) and a third bone fragment (right) (MTM V.97.38.). (D), left femur (MTM V.97.13.) in lateral; (E), medial; (F), posterior; (G), anterior; (H), proximal; (I), distal views. (J), left tibia (MTM V.97.9.) in posterior; (K), medial; (L), lateral; (M), anterior; (N), proximal; (O), distal views; (P), proximal end of fibula (MTM V.97.15.) in medial; (Q), lateral; (R), proximal views; (S), distal end of fibula (MTM V.97.43.) in lateral; (T), medial views. as, articular surface; cnc, cnemial crest; dr, dorsal rib fragment; fh, femoral head; lco, lateral condyle; mco, medial condyle; mx, matrix; pra, proximal articulation surface.

Ilium. Both ilia (MTM V.97.34., MTM V.97.44.) are preserved (Figs. 6I–6M). The left one (MTM V.97.34.) is more complete and only its medial side with the articulation of the sacral ribs is covered with sediment (Figs. 6I–6L). In general, the ilium is very similar to that of S. bollensis (MTM M 69 242) and P. typus (MTM M 62 2516) in having a rhomboidal form in lateral view with a large circular and deep acetabulum. Posteroventrally, its articulation surface for the ischium is not straight as in S. bollensis but slightly concave as that of P. typus. Dorsally, a massive and straight iliac crest is present with a pointed anterior process reaching the anterior, crested margin of the acetabulum. This process is relatively more developed than that of L. obtusidens (Johnson et al., 2017). Magyarosuchus is further similar to Pelagosaurus in having a distinct anterior acetabular flange on the ilium, created by the anterior acetabular margin. Posteriorly, the iliac crest ends in a massive triangular postacetabular process with a slightly convex posteroventral edge as in P. typus. The pubic process of the ilium can be observed only from the medial side of the right ilium that is a ventrally projected massive process, but the articulation surface is not preserved.

Ischium. Only the distal half of the left ischium (MTM V.97.36.) is preserved. It is a bard-shaped element with developed, radially oriented, elongate grooves and shallow crests on its lateral surface for muscle attachments (Fig. 6P). Whereas its ventral edge is slightly convex, the posterodorsal one is slightly concave. Its anterior process is broken. The posterior process is strongly pointed similar to that of P. typus (BSGP 1890 I 509/11, MTM M 62 2516) or S. bollensis (UH 13, Mueller-Töwe, 2006).

Pubis. Both pubes are preserved. From the right one (MTM V.97.49.) only the distal half is preserved, whereas the left one (MTM V.97.35.) is complete (Figs. 6N and 6O) but can only be studied in lateral view. It is an elongate, rod-like element with slightly widened proximal end having an oval shaped, articular surface. Whereas the posteroventral margin is almost straight, in contrast to that of P. multiscrobiculatus (SMNS 9930, Westphal, 1962), the anterodorsal one is slightly concave resulting in a widened distal end. Though a small piece of the ventral part of the distal expansion was broken off, the distal end is much wider than the proximal expansion. The lateral surface of the distal end is ornamented by radial grooves for muscle attachments. The pubis of M. fitosi differs from that of P. typus (MTM M 62 2516) in having a marked upward bending of the distal end and from that of S. bollensis where it bends rather downward (Mueller-Töwe, 2006). It also differs from the pubis of P. multiscrobiculatus in having a strongly convex distal margin.

Femur. Both femora (MTM V.97.13. left, MTM V.97.33. right) are complete but their shafts are broken and slightly dissolved due to diagenetic events (Figs. 7D–7I). In general, the femur of M. fitosi shows the typical crocodylomorph conservative shape with the shaft bending anteriorly and the proximal third of the bone curving slightly medially. The proximal end has a smooth, rounded articulation surface with dorsomedially oriented femoral head (Fig. 7H). The fourth trochanter cannot be observed. The distal end has well developed, rounded medial and lateral condyles bordering a marked intercondylar groove. The femur of M. fitosi is very similar to that of S. bollensis, P. multiscrobiculatus or P. typus, but P. multiscrobiculatus has proportionally slightly shorter femur (Westphal, 1962; Mueller-Töwe, 2006).

Tibia. Both tibiae (MTM V.97.9., MTM V.97.69.) are preserved and complete with the left one (MTM V.97.9.) being intact (Figs. 7J–7O). It is a straight, slightly anteriorly bowing element with moderately widened proximal and distal ends as seen in P. typus (MTM M 62 2516) and S. bollensis (Westphal, 1962; Mueller-Töwe, 2006). It clearly differs from the tibia of L. obtusidens in having a more gracile shaft (Johnson et al., 2017). Planes of proximal and distal ends have an angle of approximately 135° as typically seen in sauropsid tibiae. The proximal end shows two flat to very slightly concave platforms to accept the distal condyles of the femur. The cnemial crest is wide and massive projecting anteriorly. The distal end is oval shaped and slightly rounded (Fig. 7O). The length of the tibia (210 mm) is 58% of the femur length (360 mm). This proportion is more similar to that of S. bollensis (MTM M 69 242: 59%, MTM uncatalogued 1: 56%, MTM uncatalogued 2: 58%) and P. multiscrobiculatus (SMNS 9930: 60% Mueller-Töwe, 2006) than that of P. typus (MTM M 62 2516: 50%) or L. obtusidens (NHMUK PV R 3168: ∼50%).

Fibula. The two proximal halves (MTM V.97.15., MTM V.97.41., Figs. 7P–7R) and one of the distal parts (MTM V.97.43., Figs. 7S and 7T) of the fibulae are preserved. The proximal end is slightly divergent proximally and bowed in the anteroposterior plane. The proximal articular surface is oval shaped, and slightly convex. The distal end is straight and less divergent distally than the proximal end. The distal articular surface is convex and slightly obliquely oriented relative to the longitudinal axis of the bone. The preserved parts of the fibulae of M. fitosi are similar to those of S. bollensis and P. typus (Mueller-Töwe, 2006).

Astragalus. The left astragalus (MTM V.97.12.) is one of the best preserved elements of the skeleton showing all articulation surfaces (Figs. 8A–8F). It is a cubic element as typically seen in archosauromorphs (Schaeffer, 1941; Parrish, 1987; Sereno & Arucci, 1990). The dorsomedially positioned tibial articular surface is anteroposteriorly as long as mediolaterally and has a concave surface. The fibular articulation is a short block, being not as elongated as that of Simosuchus clarki (Sertich & Groenke, 2010), and the articulation surface is a well developed, tetragonal and concave surface as seen in, e.g. Proterosuchus species or in extant crocodylians (Cruickshank, 1979; Parrish, 1987; Sereno & Arucci, 1990). Dorsally the tibial and fibular articulation surfaces are separated by an anteroposteriorly short but well developed crest. Anteroventrally, the slightly convex surface is present for the metatarsal I. In posterior view, ventral to the fibular articulation, a deep groove extends lateromedially separating the dorsal part from the ventral, astragalar trochlea. This groove continues medioventrally and contains two small nutritive foramina. The astragalar trochlea ends laterally in the calcaneal peg that fitted in the socket-like articular surface of the calcaneum. This morphology indicates a ‘crocodile-normal’ (‘CN’ of Chatterjee, 1978) crurotarsal ankle type in M. fitosi, similar to that of S. bollensis, P. typus and P. multiscrobiculatus (Westphal, 1962; Mueller-Töwe, 2006). Articulation surfaces and tendon attachment areas appear to be more complex in M. fitosi than in L. obtusidens (Johnson et al., 2017). The astragali of metriorhynchids are even less complex, being mediolaterally compressed and rounded (Fraas, 1902; Andrews, 1913). The astragalus of M. fitosi was obviously freely movable relative to the tibia in contrast to that of Orthosuchus species (Nash, 1968).

Figure 8 Limb elements of Magyarosuchus fitosi gen. et sp. nov. from the Toarcian of the Gerecse Mountains, Hungary.

(A), left astragalus (MTM V.97.12.) in posterior; (B), anterior; (C), dorsal; (D), ventral; (E), lateral; (F), medial views; (G), metatarsal III (MTM V.97.10.) in proximal; (H), distal; (I), posterior; (J), lateral/medial views; (K), phalanges (MTM V.97.61.) in dorsal; (L), ventral views. (M), unidentified limb bone element (?distal end of fibula?). amt1, articulation surface for metatarsal I; cap, calcaneal peg; fia, fibular articulation surface; fo, foramen; tia, tibial articulation surface.

Metapodium. Three of the metatarsals (MTM V.97.10., MTM V.97.11., MTM V.97.45.) are preserved. Based on the length/width proportions compared to S. bollensis (MTM M 69 242) and P. typus (MTM M 62 2516), one of them (MTM V.97.10.) represents one of the third metatarsals (Figs. 8G–8J). The second specimens (MTM V.97.11.) is the distal two-third of the second or third metatarsals. They show the same morphology as the second and third metatarsals of basal thalattosuchians (Delfino & Dal Sasso, 2006; Mueller-Töwe, 2006), in having a long, straight shaft, oval to slightly rectangular cross-section, and a slightly widened distal articular end. Distal condyles are moderately developed and separated by a shallow intercondylar groove. The third specimen (MTM 97.45.) is too fragmentary to determine its more precise position.

Phalanges. A single phalanx (MTM V.97.61.) is preserved (Figs. 8K and 8L). It is two times longer than wide and has an hour-glass shape. Articular surfaces are poorly preserved. Compared to the phalanges of S. bollensis or P. multiscrobiculatus (Westphal, 1962; Mueller-Töwe, 2006), it is most similar to the first phalanx of the first manual digit of these taxa.

Unidentified limb bones. The material consists of two fragmentary limb bones. One element might represent one of the metacarpals or the ulnare (MTM V.97.38., Fig. 7C) in having a robust shaft and massive articulations proximally and distally. The other element is perhaps the distal end of the other fibula (Figs. 7S and 7T).

Dermal ossifications

Forty-three dorsal and ventral osteoderms are preserved. Many of these (MTM V.97.4., MTM V.97.24., MTM V.97.53., MTM V.97.56.) are still in matrix and only a small piece or the cross-section of them can be observed thus their position is unknown. Nevertheless, osteoderm morphology differs from those of other thalattosuchians and diagnoses M. fitosi.

Dorsal armor. Four osteoderms (MTM V.97.59., MTM V.97.60.) can be certainly referred to the dorsal armor. They are large rectangular to slightly rounded elements with an anteroposteriorly extending dorsal keel dividing the osteoderm into a greater medial and a smaller lateral part (Figs. 9A–9F). The anterior margin of the osteoderms is smooth and oblique to receive the posterior surface of the anteriorly following osteoderm. Whereas two dorsal osteoderms are flat or very slightly concave ventrally (Figs. 9A–9D), the two others are strongly bent dorsally (Figs. 9E and 9F). More prominent in these strongly bent osteoderms, but also present in the case of the two other dorsal osteoderms, is the anteroposterior keel that extends into a well-developed triangular anterolateral process (Figs. 9A, 9D and 9E). This anterolateral process is present in all basal thalattosuchians, including P. typus (as the mid-posterior dorsal osteoderms, which are typically in articulation, bear this process; MNHN.F RJN 463), but in these taxa it is more pronounced having a quite angular medial margin (Westphal, 1962) in contrast to that of M. fitosi. The shape of this process in M. fitosi is more reminiscent of the mid-posterior dorsal osteoderms of the Middle Jurassic teleosauroids L. obtusidens and Steneosaurus edwardsi (Andrews, 1913; Adams-Tresman, 1987; Johnson et al., 2017). Ornamentation is also unique in M. fitosi. Dorsal surface is ornamented by the one of the proportionally largest, irregularly shaped pits among thalattosuchians, with an extreme variation in size (from small to very large), thus the margin between the pits is frequently very thin (in some cases less than 1 mm). Pits are usually not circular but ellipsoid, bean-shaped, triangular and quadrangular in shape.

Figure 9 Osteoderms of Magyarosuchus fitosi gen. et sp. nov. from the Toarcian of the Gerecse Mountains, Hungary.

(A), dorsal osteoderm (MTM V.97.59.) in dorsal; (B), posterior; (C), posteromediodorsal views. (D), dorsal osteoderm (MTM V.97.60.) in dorsal view. (E), dorsal osteoderm (MTM V.97.60.) in dorsal; (F), posterior views. (G), block of six ventral osteoderms (MTM V.97.38.) in ventral; (H), anteroventral views. (I), block of two ventral osteoderms (MTM V.97.38.) in ventral view. aar, anterior articulation surface; alp, anterolateral process; lcr, lateral crest; su, suture.

Ventral armor. Twelve elements (MTM V.97.38.) can be referred to the ventral armor. Some of them form complex, fused blocks (Figs. 9G–9I). The largest among these contains six fragmentary osteoderms representing three–three elements of two axial rows (Fig. 9G). In these blocks the osteoderms overlap each other anteroposteriorly in a similar way as described in the dorsal osteoderms. Lateromedially, however, the osteoderms are firmly connected via interfingering sutures. Whether the ventral armor of Magyarosuchus was composed of multiple axial rows of osteoderms as seen in Pelagosaurus (Pierce & Benton, 2006) and in telosauroids (Andrews, 1913), or was built up by less axial rows, remains unclear. Ventral osteoderms are lateromedially wider than their anteroposterior length. Anteriorly they bear a smooth, oblique surface for the articulation of the overlapping anterior element. They are devoid of any processes on their margins and dorsally they do not bear crests. Their ventral surface is ornamented by large pits morphologically similar to those of the dorsal osteoderms.

Phylogenetic Analyses

Methods

Three phylogenetic analyses were conducted to assess the evolutionary relationships of M. fitosi gen. et sp. nov. within Thalattosuchia. The character scoring for M. fitosi was based on first-hand examination of the holotype by MTY, MR and AŐ. Three datasets were employed to conduct these analyses, two of which were first presented in Ristevski et al. (2018). However, both of these datasets have been extensively updated herein as they form the basis of the ongoing Crocodylomorph SuperMatrix Project. The first dataset is a merged matrix combining the two datasets originally published by Young et al. (2016), which was then subsequently revised and expanded, hereafter we refer to it as the Hastings + Young matrix (or H + Y matrix); whilst the second is an updated and expanded version of the dataset originally by Andrade et al. (2011), hereafter referred to as the modified Andrade matrix (or mA matrix). The third and final dataset used herein is that of Wilberg (2017). All data are summarised in Data S1–S5.

The first parsimony analysis presented here employs the H + Y matrix. The two parent matrices for the H + Y matrix were presented in Young et al. (2016): dataset 1 (the Hastings matrix), contained 37 operational taxonomic units (OTUs) scored for 120 morphological characters; whilst dataset 2 (the Young matrix), contained 103 OTUs scored for 298 characters. Mark Young, Alexander Hastings and Thomas Smith merged the matrices in 2016–2017. This resulted in extensive re-examination of all characters, re-scoring of characters to ensure a common and agreed philosophical approach to character construction, ensuring the OTUs from both datasets were scored for all characters, and the addition of characters from Andrade et al. (2011) and Nesbitt (2011). Some OTUs were revised and new ones added (see Ristevski et al., 2018 and T. Smith et al. 2017, unpublished data for full details). This resulted in the current iteration of the H + Y matrix containing a total of 140 OTUs scored for 454 characters. Excluding M. fitosi, seven of the 140 OTUs are basal metriorhynchoids, 42 are metriorhynchids, and 18 are teleosauroids. A total of 25 characters representing morphoclines were treated as ordered (7, 28, 36, 49, 57, 98, 164, 166, 174, 205, 225, 228, 234, 264, 274, 330, 357, 362, 372, 407, 410, 420, 421, 423, 435). For the H + Y matrix, Postosuchus kirkpatricki Chatterjee, 1985 was used as the outgroup taxon.

The second parsimony analysis presented here employs the mA matrix: a modified version of the character and taxon list first published by Andrade et al. (2011), which originally included 104 OTUs scored for 486 characters. As per the recommendations of Andrade et al. (2011), Halliday et al. (2015), and Puértolas-Pascual, Canudo & Sender (2015), the putative goniopholidid Denazinosuchus kirtlandicus (Lucas & Sullivan, 2003), and the Asian taxon ‘Goniopholis’ phuwiangensis Buffetaut & Ingavat, 1983 OTUs, along with the composite ‘Goniopholis’ phuwiangensis + Siamosuchus terminal (ALTSiamosuchus), were excluded due to their instability and, in the case of the latter, inapplicability. Following Halliday et al. (2015) and Ristevski et al. (2018) the putative goniopholidids Kansajasuchus extensus Efimov, 1975, Sunosuchus shartegensis Efimov, 1988 and Turanosuchus aralensis Efimov, 1988 were excluded due to their instability. In total, the analysis of the mA matrix presented here included 110 OTUs scored for 570 characters. Excluding M. fitosi, one of the 110 OTUs are basal metriorhynchoids, 10 are metriorhynchids, and three are teleosauroids. Thirty-one characters representing morphoclines were treated as ordered (6, 9, 32, 71, 72, 125, 146, 153, 158, 216, 218, 222, 245, 271, 297, 302, 303, 326, 355, 378, 379, 446, 467, 471, 481, 523, 526, 536, 537, 539, 551). For the mA matrix, Gracilisuchus stipanicicorum Romer, 1972 was the outgroup taxon.

For both the H + Y and mA datasets the primary differences between our analyses and those presented by Ristevski et al. (2018) are: (1) the continued merging of the two datasets as part of the Crocodylomorph SuperMatrix Project, which has been the primary cause of the general increase in character number of both datasets; (2) the addition of M. fitosi into both datasets; (3) revision of current characters based on those from Nesbitt (2011), Narváez et al. (2015), Buscalioni (2017), Leardi, Pol & Clark (2017) and Nesbitt & Desojo (2017), and the addition of new ones from those papers; and (4) the creation of new characters to help explore some morphofunctional complexes (such as the hypocercal tail in metriorhynchoids).

The Wilberg dataset (hereafter W matrix), is largely the same as that presented in Wilberg (2017). The only differences are: (1) the addition of M. fitosi; (2) the addition of two characters from the two merged datasets (the mandibular parallel surangulodentary and angulodentary grooves character and the flange-like ilium anterior margin character); and (3) some minor rescoring of teleosauroids based on personal observations by MTY (see Online Supplementary). This resulted in the current iteration of the W matrix containing a total of 98 OTUs scored for 408 characters. Excluding M. fitosi, five of the 98 OTUs are basal metriorhynchoids, 12 are metriorhynchids, and 10 are teleosauroids. A total of 40 characters representing morphoclines were treated as ordered (26, 51, 58, 59, 61, 64, 83, 128, 148, 151, 163, 202, 203, 208, 210, 220, 224, 240, 255, 261, 263, 265, 270, 296, 304, 311, 316, 342, 344, 345, 346, 362, 366, 373, 375, 384, 386, 393, 394, 399). For the W matrix, G. stipanicicorum was used as the outgroup taxon.

The cladistic analyses were conducted following the methodology implemented by Young et al. (2016), using TNT v1.5, Willi Hennig Society Edition (Goloboff & Catalano, 2016). Memory settings were increased with General RAM set to 900 Mb and the maximum number of trees to be held set to 99,999. In the analysis of each matrix, cladogram space was searched using the advanced search methods in TNT (sectorial search, ratchet, drift and tree fusion) for 1,000 random addition replicates. The default settings of the advanced search methods were modified to increase the number of iterations of each method per analysis replicate (except for tree fusion, which was kept at three rounds). For the sectorial search, 1,000 drifting cycles were applied for selections of above 75 with 1,000 starts, and trees were fused 1,000 times for those below 75. TNT also conducted 1,000 rounds of consensus sectorial searches (CSS) and 1,000 rounds of exclusive sectorial searches (XSS). The analysis included 1,000 ratchet iterations with the cease perturbation phase reached when 1,000 substitutions were made or 99% of swapping was completed. The program incorporated 1,000 drift cycles within the analysis, which also reached the cease perturbation phase at 1,000 substitutions made or 99% of swapping completed.

Results

The first phylogenetic analysis that utilised the H + Y matrix recovered 84 most-parsimonious cladograms (MPCs) with 1,477 steps (ensemble consistency index, CI = 0.417; ensemble retention index RI = 0.842; rescaled consistency index RC = 0.351; ensemble homoplasy index HI = 0.583). Overall, the strict consensus topology recovered from this analysis (Fig. 10A) is very similar to the ones presented in Ristevski et al. (2018) and T. Smith et al., 2017, unpublished data. The only difference within Thalattosuchia is the addition of M. fitosi, which is found to be the sister taxon of P. typus (Fig. 10A). The overall picture of crocodylomorph interrelationships found herein are the same as those found in previous iterations of this merged dataset (Ristevski et al., 2018; T. Smith et al. 2017, unpublished data): ‘sphenosuchians’ form a grade and Protosuchidae and the shartegosuchid Fruitachampsa callisoni Clark, 2011 are recovered as basal crocodyliforms. The remaining taxa comprise Mesoeucrocodylia, which includes a clade formed by Eopneumatosuchus colberti Crompton & Smith, 1980 + Thalattosuchia, and the other clade being Metasuchia. Metasuchia contains two sub-clades, Notosuchia and Neosuchia. Within Thalattosuchia, both Teleosauridae and Metriorhynchoidea are recovered as monophyletic. P. typus is found to be a basal metriorhynchoid, and Metriorhynchidae, Metriorhynchinae, Rhacheosaurini, Geosaurinae and Geosaurini are all found to be monophyletic (Fig. 10A).

Figure 10 Results of the phylogenetic analyses.

(A), Strict consensus of 16 most parsimonious cladograms based on the modified Andrade matrix (Andrade et al., 2011), showing the phylogenetic relationships of Magyarosuchus fitosi gen. et sp. nov. within Metriorhynchoidea. (B), Strict consensus of 84 most parsimonious cladograms based on the Hastings + Young matrix (Young et al., 2016), showing the phylogenetic relationships of Magyarosuchus fitosi gen. et sp. nov. within Metriorhynchoidea.

The second phylogenetic analysis that utilised the mA matrix yielded 16 MPCs with 2,472 steps (CI = 0.305; RI = 0.764; RC = 0.233; HI = 0.695). Overall, the strict consensus topology recovered from this analysis is very similar to the ones presented in Ristevski et al. (2018) and T. Smith et al. 2017, unpublished data. The only differences within Thalattosuchia are: (1) the addition of M. fitosi, which is found to be the sister taxon of P. typus; (2) teleosauroids are no longer in a polytomy, but now S. bollensis is the sister taxon to a clade S. heberti + P. multiscrobiculatus; and (3) Metriorhynchus superciliosus is no longer the sister taxon to Geosaurini (but in a trichotomy with Geosaurini and Rhacheosaurini) (Fig. 10B). The overall picture of crocodylomorph interrelationships found herein are the same as those found in previous iterations of this dataset: ‘sphenosuchians’ form a grade and Protosuchidae, Gobiosuchus and Hsisosuchus are recovered as successively more derived basal crocodyliforms. The remaining taxa comprise Mesoeucrocodylia, which contains two sub-clades, Notosuchia and Neosuchia. Thalattosuchia is recovered within Neosuchia, as the sister taxon to Tethysuchia. Within Thalattosuchia, both Teleosauridae and Metriorhynchoidea are recovered as monophyletic. P. typus is found to be a basal metriorhynchoid, and Metriorhynchidae, Rhacheosaurini and Geosaurini are all found to be monophyletic (Fig. 10B).

The final phylogenetic analysis, utilising the W matrix, recovered six MPCs with 1,777 steps (CI = 0.306; RI = 0.733; RC = 0.224; HI = 0.694). The strict consensus topology recovered from this analysis is almost identical to the analysis by Wilberg (2017), the only difference is the addition of M. fitosi (which is found to be the sister taxon of P. typus). The overall picture of crocodylomorph interrelationships found herein are the same as that found in Wilberg (2017): with ‘sphenosuchians’ forming a grade and thalattosuchians being sister taxon to Crocodyliformes (Fig. 11). Within Crocodyliformes there are two sub-clades: Mesoeucrocodylia, and one formed by Protosuchidae, Shartegosuchidae, Gobiosuchidae and Hsiosuchus. The remaining taxa comprise Mesoeucrocodylia, which contains two sub-clades, Notosuchia and Neosuchia. Within Thalattosuchia, both Teleosauroidea and Metriorhynchoidea are recovered as monophyletic. P. typus is found to be a basal metriorhynchoid, and Metriorhynchidae, Geosaurinae and Geosaurini are all found to be monophyletic (Fig. 11).

Figure 11 Strict consensus of six most parsimonious cladograms based on the Wilberg matrix (Wilberg, 2017), showing phylogenetic relationships of Magyarosuchus fitosi gen. et sp. nov. within Metriorhynchoidea.

Although the three phylogenetic analyses do not recover Thalattosuchia in the same region of the crocodylomorph tree, there are many aspects they do agree upon: The monophyly of Thalattosuchia.

The separation of Thalattosuchia into two clades: Teleosauroidea and Metriorhynchoidea.

That P. typus is a basal metriorhynchoid.

The sister group relationship between P. typus and M. fitosi, which forms the basal-most sub-clade of Metriorhynchoidea.

The monophyly of Metriorhynchidae.

The monophyly of Geosaurini.

This suggests that the newer, larger, phylogenetic datasets being compiled on thalattosuchian internal relationships are becoming less sensitive to where in Crocodylomorpha Thalattosuchia is recovered (see also Jouve, 2009; Wilberg, 2015b). Although all three datasets do have interesting internal differences in the arrangement of Teleosauroidea and the monophyly or not of Metriorhynchinae, there is a growing consensus between them. The recovery of Steneosaurus gracilirostris as the basal-most teleosauroid in the H + Y and W matrices (Figs. 10A and 11) is especially interesting, as it polarises laterally oriented orbits as being symplesiomorphic for Thalattosuchia (with the dorsal orientation being a convergence between derived teleosauroids and neosuchians).

Given that, except for P. typus, the postcranial anatomy of basal metriorhynchoids is poorly known, we tested whether this species is the sole responsible taxon for pulling the mostly postcranial-based M. fitosi among basal metriorhynchoids. The exclusion of P. typus from the mA matrix retains M. fitosi at the base of Metriorhynchoidea. Excluding P. typus from the W and H + Y matrices finds M. fitosi close but unresolved relative to other basal metriorhynchoids (in the case of the H + Y matrix only when the highly fragmentary Peipehsuchus teleorhinus is removed from the consensus tree) although few of the alternative positions are supported by synapomorphies. The absence of common synapomorphies is due to the lack of P. typus and the inclusion of few basal metriorhynchoids in the H + Y/W matrices, all of which lack post-crania and therefore cannot be commonly scored for the post-cranial characters that could unite M. fitosi with other metriorhynchoids. The metriorhynchoid affinity of M. fitosi is therefore rather reasonable (e.g., presence of enlarged femoral medial tuber; coracoid with convex proximal and distal ends; oval-shaped sacral vertebral centrum) but we cannot exclude that it may be more derived within the group because little is known about character evolution at the base of the clade.

Discussion

Thalattosuchian marine adaptations

Postcranial elements in basal thalattosuchians (especially in metriorhynchoids Young et al., 2010) are poorly known, thus the early phases of their adaptation to a fully aquatic lifestyle is still speculative. Wilberg (2015a) listed a number of skeletal adaptations thought to be linked to an increasingly marine lifestyle in thalattosuchians, such as: (1) the reorientation of the orbit from dorsal to laterally directed (Hua & Buffrénil, 1996), (2) development of hypertrophied nasal exocrine glands (Fernández & Gasparini, 2008; Gandola et al., 2006), (3) humerus mediolateral flattening and a reduction in diaphysis length (both in Teleosauroidea and Metriorhynchoidea), (4) reduction of relative tibia and ulna length, (5) reduction and loss of osteoderm cover, (6) modification of the pelvis, (7) development of a hypocercal tail with a distinct regionalisation of the distal caudal vertebrae (Fraas, 1902; Andrews, 1913; Hua & Buffetaut, 1997; Young et al., 2010).

Magyarosuchus sheds new light on the early evolutionary history of marine adaptations in Thalattosuchia. Most of the elements in Magyarosuchus seem to indicate a body-plan similar to basal teleosauroids: in having elongated limb bone diaphyses with well-developed proximal and distal epiphyses, a ‘primitive’ pelvis construction (robust iliac peduncles, retention of iliac postacetabular process), and the presence of complex and heavy dorsal and ventral osteoderm cover. The astragalus is very complex with well-developed articulation surfaces for the tibia, fibula and metatarsal I, and with the presence of the calcaneal peg it shows the typical ‘crocodile-normal’ (‘CN’ of Chatterjee, 1978) crurotarsal ankle joint. These features suggest that adaptation to marine habitats in Magyarosuchus could have been similar to that of the Early Jurassic teleosauroids S. bollensis, ‘Steneosaurus’ gracilirostris and P. multiscrobiculatus (Westphal, 1962).

One caudal vertebra (Figs. 5K–5P), however, reveals some features that are not present in these teleosauroids or in basal metriorhynchoids. This vertebra is the smallest and distal most element (Figs. 5K–5P and 12I) among the preserved caudals. According to the proportion of vertebral centrum height between the dorsal vertebrae and distal caudal vertebrae measured in P. typus (MTM M 62 2516), the distal-most preserved caudal of M. fitosi represents one of the last 10–15 elements in the caudal series. In S. bollensis (MTM M 69 242; Westphal, 1962) and P. typus (MTM V.52.2516), these caudals have only reduced, anteroposteriorly short, and slightly posteriorly projected neural spines (Figs. 12E–12H). The small caudal of Magyarosuchus, on the other hand, possesses an anteroposteriorly long and dorsally projecting, elongate neural spine (Fig. 12I). Although the dorsal end of the neural spine and the anterior end of the centrum is missing (Figs. 5N and 5O), it clearly differs from the distal-most vertebrae of basal teleosauroids or P. typus. We suggest that this vertebra represents the bending zone of the distal end of the caudal series to strengthen a slight tail fin. Tail fins are present, e.g. in the metriorhynchids M. superciliosus (GPIT RE 9405), ‘Metriorhynchus’ brachyrhynchus (NHMUK PV R 3804), Gracilineustes leedsi (NHMUK PV R 3014), Rhacheosauus gracilis (NHMUK PV R 3948) and Cricosaurus suevicus (SMNS 9808). An isolated bending zone caudal vertebra is also known for Torvoneustes carpenteri (Wilkinson, Young & Benton, 2008). In these forms three to four vertebrae of the bending zone have at least two to three times longer neural spines than the previous caudals and the centra are slightly bent with shorter ventral margin (Fraas, 1902; Andrews, 1913). The small caudal of M. fitosi is missing its anterior part, but, based on the shape of the posterior articulation of the centrum it might have not been as bended as that, e.g. in M. superciliosus. It seems that in M. fitosi the distal tail was still not as ventrally deflected as in metriorhynchids; the neural spines, however, became elongated to stiffen at least a rudimentary caudal fin. Moreover, these bending zone caudals in metriorhynchids (and M. fitosi) have a centrum that is mediolaterally compressed relative to the pre-bending vertebrae.

Figure 12 Comparison of thalattosuchian bony tails and the distal caudal vertebrae within the bending zone.

(A and B), Cricosaurus suevicus from Nusplingen (GPIT RE 7322); (C and D), Metriorhynchus superciliosus (GPIT RE 9405); (E and F), Steneosaurus bollensis (MTM M 69242); (G and H), Pelagosaurus typus (MTM M 62 2516); (I), Magyarosuchus fitosi gen. et sp. nov. distal caudal (MTM V.97.19.) with the interpreted original outline of the vertebra.

This remarkable feature fits well with the mosaic evolution of marine adaptations in thalattosuchians proposed by Wilberg (2015a). Since the skull is unknown in M. fitosi, no other skeletal modifications refers to a pelagic habit in this form, except for this modified distal caudal. This suggests that a caudal fin supported by a ventrally bended row of distal caudals and a few distal caudals with elongated neural spines should have occurred by the later part of the Early Jurassic, much earlier in thalattosuchian history than the presently available record shows (later part of the Middle Jurassic, Callovian; Young et al., 2010).

Body length

As there is no complete skull, the only metric to establish a body length estimate was femoral length. However, based on teleosauroids, Young et al. (2016) found femoral length to be the more reliable metric for estimating total length of those thalattosuchians. Both femora of M. fitosi are broken and partial dissolved. Taking the raw measurements of the femora and using the femoral length vs. body length equations of Young et al. (2011, 2016) we get a range of body length values: 4.6–4.8 m. This is based on: (1) difference in size between the left and right femora due to preservation, and (2) the uncertainty of whether to use the metriorhynchid equation from Young et al. (2011) or the two teleosauroids equations of Young et al. (2016). (Note that Young et al. (2016) had two equations: first based on a complete skeleton sample of 12, and a slightly larger sample of 16 with added some less complete skeletons.)

If we assume M. fitosi had a scaling ratio similar to teleosauroids, and only use the more complete right femur, this yields a body length estimate of 4.67–4.74 m. However, if M. fitosi had a scaling ratio similar to metriorhynchids, and we only use the more complete right femur, this gives a body length estimate of 4.83 m. Interestingly, Young et al. (2016) found using the metriorhynchid body length equations to more reliably estimate the size of two P. typus skeletons. This suggests that basal metriorhynchoids may have had a scaling ratio more similar to metriorhynchids than teleosauroids. However, as the sample was only of two P. typus specimens this conclusion remains untested.

Regardless of which equation is correct, a body length of 4.67–4.83 m makes M. fitosi the largest known non-metriorhynchid metriorhynchoid. It is substantially larger than the only other Early Jurassic metriorhynchoid P. typus, which is typically 2–3 m in length. Furthermore, the fragmentary material of other basal metriorhynchoids all suggest taxa closer in size to P. typus than M. fitosi, or perhaps reaching 3.5 m (see Eudes-Deslongchamps, 1867–1869; Collot, 1905; Mercier, 1933; Wilberg, 2015a; NHMUK PV R 2681, NHMUK PV R 3353). Moreover, these length estimates also mean M. fitosi was larger than most metriorhynchid specimens estimated by Young et al. (2011), as few metriorhynchid species exceeded 4.5 m in length, and those that did were the larger-bodied macrophagous taxa.

Compared to known Early Jurassic teleosauroids, M. fitosi was within the size range of the larger-bodied species. Few Early Jurassic thalattosuchians are known to exceed 4.5 m, with species such as P. multiscrobiculatus and ‘Steneosaurus’ gracilirostris typically in the 2–3 m range (see Westphal, 1962; NHMUK PV OR 14792, SMNS 9930). The holotype of ‘Steneosaurus’ brevior (NHMUK PV OR 14781) is that of a large skull and lower jaw, with an approximate length of 88.3 cm. Using the cranial to body length equations of Young et al. (2016), it has an estimated body length of 4.47–4.58 m. However, there are specimens, which albeit are rare, of S. bollensis reaching, and even exceeding, 5 m (see Westphal, 1962; Young et al., 2016). Therefore, the largest Early Jurassic thalattosuchians, and crocodylomorphs, were most likely teleosauroids. This trend continues into the Middle Jurassic and on into the Early Cretaceous with teleosauroids reaching greater body lengths than metriorhynchoids (see Young et al., 2016).

Conclusion

Here, we describe a new crocodylomorph taxon, M. fitosi get. et sp. nov., based on a new skeleton from the Gerecse mountains of Hungary. Despite being incomplete and lacking the cranium, we demonstrate that this late Lower Jurassic taxon shows remarkable similarities with the iconic Lower Jurassic genus Pelagosaurus. Magyarosuchus and Pelagosaurus are found to be sister taxa in all three phylogenetic analyses undertaken herein, although the two characters uniting this arrangement are not known from other basal metriorhynchoids (due to poor preservation of taxa such as Teleidosaurus, Eoneustes and Zoneait). Therefore, we cannot be certain that the sister relationship between Magyarosuchus and Pelagosaurus is natural, or due to incomplete information. Regardless, both are found to be basal metriorhynchoids, near the start of the radiation that yielded dolphin-like crocodyliforms. Interestingly, M. fitosi is the oldest known thalattosuchian discovered from an ‘ammonitico rosso’ type pelagic deposit (rather than the usual estuarine, lagoonal or coastal ecosystems Lower Jurassic thalattosuchians are discovered from). The pelagic depositional environment and neritic associated cephalopod fauna are both consistent with the inferred open-marine adaptation of M. fitosi, namely a mediolaterally compressed distal caudal vertebra with an unusually elongated and dorsally projected neural spine which suggests the presence of a distal tail structure that could have been a hypocercal fin, or a precursor to it. The unique combination of retaining heavy dorsal and ventral armor, while having a slight hypocercal tail, on the other hand, highlights the mosaic manner of marine adaptations in Metriorhynchoidea. Furthermore, it underscores how little is still known about the timing and tempo of metriorhynchoid pelagic adaptations and their early radiation.

Supplemental Information

Supplemental Information 1 Charaters used in the phylogenetic analyses.

Click here for additional data file.

Supplemental Information 2 Taxon-character matrices used in this study.

Click here for additional data file.

Supplemental Information 3 Analysis of data 1.

Click here for additional data file.

Supplemental Information 4 Analysis of data 2.

Click here for additional data file.

Supplemental Information 5 Analysis of data 3.

Click here for additional data file.

We thank Attila Fitos, discoverer of the specimen and Z. Sirányi, Z. Szabó and I. Szabó for the excavation and early preparation of the specimen. We are grateful to P. Gulyás, R. Kalmár for preparation and M. Szabó (ELTE) for taking photographs of the specimens, and János Magyar for technical help. We thank L. Kordos (Budapest, Hungary) for his advises given during the project and his comments to an earlier version of the manuscript. We are grateful to Yanina Herrera and an anonymous reviewer for their constructive comments that highly improved our manuscript. Mark T. Young would like to thank R. Allain (MNHN), E. Maxwell and R. Schoch (SMNS), D. Schwarz (MfN), D. Vasilyan and I. Werneburg (GPIT), and M. Gasparik and Z. Szentesi (MTM) for collections access during these trips.

Institutional Abbreviations

BRLSI M Moore Collection of the, Bath Royal Literary and Scientific Institute, Bath, UK

BSPG Bayerische Staatssammlung für Paläontologie und Historische Geologie, München, Germany

GPIT Paläontologische Sammlung der Eberhard Karls Universität Tübingen, Tübingen, Germany

MNHN.F Fossil Collection of the Muséum National d’Histoire Naturelle, Paris, France

MTM Hungarian Natural History Museum, Budapest, Hungary

NHMUK PV Vertebrate palaeontology collection of the Natural History Museum, London, UK (OR, old register; R, reptiles)

SMNS Staatliches Museum für Naturkunde, Stuttgart, Baden-Württemberg, Germany

UH Urweltmuseum Hauff Holzmaden.

Additional Information and Declarations

Competing Interests

Author Contributions

Data Availability

New Species Registration

Mark T. Young is an Academic Editor for PeerJ.

Attila Ősi analysed the data, contributed reagents/materials/analysis tools, prepared figures and/or tables, authored or reviewed drafts of the paper, approved the final draft.

Mark T. Young analysed the data, contributed reagents/materials/analysis tools, prepared figures and/or tables, authored or reviewed drafts of the paper, approved the final draft.

András Galácz analysed the data, authored or reviewed drafts of the paper, approved the final draft.

Márton Rabi analysed the data, contributed reagents/materials/analysis tools, prepared figures and/or tables, authored or reviewed drafts of the paper, approved the final draft.

The following information was supplied regarding data availability:

The raw data are provided in Supplemental Dataset Files.

The following information was supplied regarding the registration of a newly described species:

Publication LSID: urn:lsid:zoobank.org:pub:3623D096-C737-4B69-A491-ABC0F50FF4D4

Genus name: urn:lsid:zoobank.org:act:CA3C4B54-B457-4970-8A99-8730935F17FF

Species name: urn:lsid:zoobank.org:act:E87FBEBB-EE13-4415-A226-1D2FA16CAD24.

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
