# Peer review of "A new large-bodied thalattosuchian crocodyliform from the Lower Jurassic (Toarcian) of Hungary, with further evidence of the mosaic acquisition of marine adaptations in Metriorhynchoidea"

_PeerJ, doi:10.7717/peerj.4668_

## Round 0.1 · original submission · Minor Revisions

Both reviewers offer a number of helpful suggestions, which should be addressed in preparing the revised version of the manuscript. In general, the submission looks fine.

Reviewer 1 ·

Basic reporting

no comment

Experimental design

no comment

Validity of the findings

no comment

Additional comments

Ösi et al describe a new species of metriorhynchoid thalattosuchian from the Lower Jurassic of Hungary. While highly fragmentary, the new specimen contains enough information to be diagnosable, fills in some much-needed information on postcrania of non-metriorhynchid metriorhynchoids, and represents the earliest evidence of thalattosuchians in the pelagic realm. I think this is a generally well written and executed manuscript. I have a number of minor issues and formatting suggestions, but nothing that should unduly delay its publication.

Comments:
Pelagosaurus typus - the authors typically write out the full genus and species name when discussing this species. However, in some places it is abbreviated as Pe. typus, and in others as P. typus (e.g. towards the end of the phylogenetic results section). I'm assuming the use of "Pe. typus" was meant to clarify as Platysuchus is also discussed extensively (and Peipehsuchus occasionally). The authors should be consistent.

Lines 73-79: This entire paragraph seems more like discussion to me rather than introduction. It presents and interprets results of analyses the paper hasn't yet described. This paragraph actually sounds more like a "conclusions" paragraph than anything. I recommend this be removed from the introduction. This will probably require writing a new final paragraph for the introduction. Otherwise it will end very abruptly.

Lines 134-135: "A very rough sketch…" - I'm not sure why this sentence was included as it seems irrelevant.

Line 138: "preserved in laminated limestone" - These specimens are preserved in shale, rather than limestone assuming the authors mean the incredibly numerous specimens from the Holzmaden region of Germany (Posidonia Shale).

Lines 224-225: "distinct anterior acetabular flange…" - It would be nice to see side by side comparisons of this feature of the ilium of Magyarosuchus and Pelagosaurus. Also for the surangulodentary and angulodentary grooves. Perhaps a figure showing these could be added (in the supplement would be fine if there isn't room in the manuscript for more figures).

Line 241: "anteroposteriorly extending grooves…" you name these grooves in the diagnosis (surangulodentary and angulodentary) - they should be named here as well.

Line 254: "… form the shallow lateral side of the coronoid process" - ? The dentary doesn't typically contribute to the lateral side of the coronoid process in other thalattosuchians.

Lines 260-261: "shelf is present that has a slightly elevating medial side" - could this shelf be an articulation surface for the surangular?

Lines 262-271: In the figure for this specimen, the authors have labeled the external mandibular fenestra (or where it would be if the specimen weren't crushed), however they make no mention of that in the text here. This should be discussed as this is a feature lost in metriorhynchids. They should also discuss the deformation that has closed the EMF to explain why it is not visible in the figure.

Lines 272-273: I agree it is too fragmentary to describe in detail. However, is there any surface ornamentation visible? It looks like there might be from the figure, but it's hard to tell. If there is ornamentation, this should be mentioned as you discuss the ornamentation of the other mandibular pieces.

Lines 291-292: "are devoid of pneumatic foramina" - While true, I don't think this bears mentioning - I don't know of any crocs with pneumatized vertebrae.

Line 292: "neural arches are fully fused to the centra…" - this is typically used as a measure of ontogenetic stage (e.g. Brochu 1996; Irmis 2007). The authors may wish to note that the closure of these sutures support that this was a very mature individual (in agreement with the large size).

Line 300: "lumbar" is typically a vertebral classification for mammals (and their relatives). I would use "dorsal" here.

Lines 309-310: "… neural arches seem to be fused, the neural spines are separated" - I'm not sure what the authors mean here. Are they saying that the centra are unfused, the neural spines are separate, but the zygopophyseal articulations are fused? More clarity would be helpful here.

Lines 336-337: description of sacral ribs - The authors may want to mention here how the sacral ribs of this specimen differ from the elongate, slender sacral ribs of metriorhynchids (but are similar to those of Pelagosaurus).

Line 339: "probably represents the first sacral rib" - does its articulation surface match that of S1?

Line 363: Is this "triangular boss" the same thing as the postacetabular process (as labeled in Fig. 6)?

Line 366: "bard-shaped" - I don't know what this means.

Line 377: "a widened distal end" - Perhaps emphasize that the distal end is much wider than the proximal expansion. Also, is the ventral part of the distal expansion broken off? Is the expansion asymmetrical (i.e. is the dorsal portion more expanded than the ventral?)?

Line 435: "is the two-third of the second or third metatarsals" - I'm not sure what this is trying to say. Is this specimen the distal two thirds of one of these metatarsals? The sentence might just be missing a couple of words.

Line 448: Fig. 6C - Do the authors mean 7C? Figure 6C is a sacral rib.

Lines 460-461: "anterior margin of the osteoderms… overlap the posterior surface of the anteriorly following osteoderm" - This sounds backwards to me (though the wording is a bit confusing). The posterior margin of each osteoderm should overlap the anterior margin (and anterolateral process) of the following osteoderm.

Lines 472-473: "ornamented by the proportionally largest pits among thalattosuchians" - did the authors quantify this? It doesn't look noticeably different from some other specimens I've seen (e.g. some S. bollensis, Teleosaurus, or Pelagosaurus osteoderms). Maybe a figure of side by side comparisons would help?

Lines 475-481: The authors don't make any mention of the ventral osteoderms being sutured to one another, though this is indicated in the figure.

Line 489: "are also in an 'in review' manuscript" - As these datasets have already been published in Ristevski et al., I'm not sure there's any need to reference an 'in review' manuscript.

Lines 572-573: "Postosuchus kirkpatricki lies outside the clade…" - I don't think this statement is necessary. The authors rooted their tree on Postosuchus, so this relationship is forced by your input parameters. The same goes for the other two analyses rooted on Gracilisuchus. Gracilisuchus lying outside of Crocodylomorpha was forced a priori by designating it as the outgroup.

Lines 625-626: "internal relationships are becoming less sensitive to where in Crocodylomorpha Thalattosuchia is recovered" - The issue of thalattosuchian relationships changing based on location within the larger crocodylomorph tree were discussed previously by Jouve (2009 - the Teleosaurus description) and Wilberg (2015 - the outgroup paper).

Lines 634-635: Magyarosuchus is referred to twice here as "Magyarosaurus". The authors should check the rest of the manuscript to ensure that there are no other references to "Magyarosaurus".

Lines 672-673: "some features still unknown in these…" - These features aren't really "unknown" in S. bollensis or Pelagosaurus as complete caudal series are known for both. I think the authors mean that this Magyarosuchus caudal vertebra possesses a morphology different from that of teleosaurs and Pelagosaurus.

Line 727: "P. typus, which is typically 2-3m in length" - unimportant side note: there is a very large Pelagosaurus skull at the Stuttgart museum. The specimen number is kind of hard to read in my photographs, but I think it is "SMNS 91.102" - it is labeled as "unusually large skull". Just eyeballing it from my pictures, the total body length should be around 4m based on the cranial/body length regressions in Young 2016, which is still smaller than your estimates for Magyarosuchus.

I didn't look over the References section in any detail, but there appear to be at least a few minor issues (e.g. the Hua and Buffrénil 1996 citation is partially in bold, and "vertebrate" is misspelled). Some references include both the journal issue and volume, while others only include the issue. The authors should check over their references for consistency.

Minor grammatical issues:

Lines 90-91: Urweltmuseum Hauff is in a different font than the rest of the manuscript.

Line 94: the word "the" should precede "specimen" in this sentence.

Line 147: the word "A" should precede "vibro-tool".

Lines 181-194: some, but not all, specimen numbers end in a period. Is this intentional?

Line 199: "for thanking his donation" - this doesn't seem like the right wording. Maybe "in thanks for his donation"?

Line 223: I think the word "rami" is unnecessary here.

Line 280: the word "the" should precede "crown surface"

Line 304: I would use the word "presumably" rather than "supposedly" in this context.

Line 372: "is only the distal half preserved" - should be "only the distal half is preserved"

Line 442: "precize" should be "precise"

Line 479: "avoid" should be "devoid"

Line 483: "Analysis" should be "Analyses"

Line 640: There is an unnecessary line break in the middle of this line.

Line 739: "questions" should be "equations"

Line 762: "usually" should be "unusually"

Comments on figures:
Figure 3 caption - "dentale" should be "dentary" throughout
- "… compressed to each other preventing to outline" - this is odd wording. Maybe change to "the specimen is dorsoventrally compressed, artificially closing the external mandibular fenestra", or something like that.

Figure 3 - the grooves visible in A are given specific names in the diagnosis (surangulodentary and angulodentary) but are just called "grooves" on this figure. I think they should be labeled more specifically on the figure (or at least. In the figure caption). The authors could leave the "gr" label on the figure and just change the figure caption to indicate that "gr" = surangulodentary and angulodentary grooves.
- I think it would help in interpreting these dentary fragments if the alveoli were outlined (maybe with a translucent gray oval). Some of their locations are labeled, but I think they would be clearer with some sort of emphasis. I also think some lines indicating sutures or contacts between bones would be helpful. The angular and surangular are labeled on G, but their contacts aren't that clear.
- There are some lines pointing to something on H and I without a label.

Figure 4 caption - Are there specimen numbers for the teeth shown in A, B, and G? The tooth in C has its own number.
- Which teeth are shown in H-I? Are these just magnified views of the tooth in G or two different teeth?

Figure 4 - in part I, the carina is somewhat out of focus near the top of the image. Would it be possible to retake this photo with a greater depth of field? Also, there is a scale bar in this image, but it is not indicated here or in the figure caption what the scale represents (mm?).

Figure 5 - It is somewhat difficult to determine which images are of the same vertebra(e) in different views (e.g., on first impression, I would assume H is ventral view of the vertebra shown in I and J). Maybe it would be helpful to box them off (i.e. box around A-C, D-C, F-H, I-J, and K-P). A similar argument could be made for figures 6-9, but these are generally more obvious.

Figure 10 - The caption refers to the trees as A and B, but they are labeled "Hastings & Young matrix" and "modified Andrade matrix". Also, the trees are mislabeled. The one labeled Hastings and Young is the Andrade tree, and vice versa.
- On both of these trees "GONIOPHOLIDAE" should be "GONIOPHOLIDIDAE"
- There is something off with the font in these figures (and in Fig. 11). It looks like some of the text has been compressed (top to bottom), and some has been stretched when the authors were resizing the trees.
- Minor points: I'm not sure I like the gray ovals behind the trees. The darker one behind the lower tree especially reduces the contrast between the black text and the background making the words more difficult to read. If the authors wish to retain the ovals as a way to obviously separate the trees, perhaps they could lighten up the gray color. It doesn't have to be very dark to set it apart from the white of the page.

Figure 12 caption - Magyarosaurus should be Magyarosuchus.

Table 1 - "dentale" should be "dentary" throughout. "coracoideum" should be "coracoid"

Comments on supplementary files:
Data file S1 - There are still some track changes comments in here.
- Some formatting issues in tables (particularly S5.2). Also, the character # column is too narrow starting with 425 of H+Y matrix
- Parts S4 and S6 appear to be identical (the Wilberg matrix stuff)
- Citation for von Meyer 1831 has a weird giant "2" next to it (Wilberg, 2015a has a giant "4")

Data file S2 seems to be an Excel format version of the Wilberg matrix, which is also provided as a nexus file in data file S6. Probably only the nexus file is necessary.

Data file S3 is also the Wilberg matrix info already included in S1.

I had an issue opening data file S4 (I assume the H+Y matrix?) - Mesquite gave the error message "There was a problem reading the character matrix Character Matrix. It appears that the file is corrupt ( taxon 140, character 17 (section of matrix as stored: "osaur ????"))."
- Thus, I was unable to double check the results of this analysis (though I don't expect there to be any issues). The authors should check this file.

·

Basic reporting

The manuscript “A new large-bodied thalattosuchian crocodyliform from the Lower Jurassic (Toarcian) of Hungary, with further evidence of the mosaic acquisition of marine adaptations in Metriorhynchoidea” will be an interesting contribution to the knowledge of basal thalattosuchians. The manuscript is well organized and writing is clear. The figures of the specimen are clear. Nevertheless, some modifications are needed in order to be accepted for publication.
Some comments are highlighted in the pdf.

Experimental design

no comment

Validity of the findings

no comment

Additional comments

Diagnosis:
I not convinced about the use of the body length as a diagnostic feature. May be the authors can start the sentence as follow: Large size metriorhynchoid thalattosuchian with the following unique combination ….. and delete the sentence “large body size (estimated body length: in the range of 4.67–4.83 m)” from the diagnosis.

Institutional abbreviations:
Lack the name of the institutions that correspond to the acronyms BSPG and MNHN.

Description and Comparisons:
Comparisons of postcranial elements need to be improved. It would be nice to see more comparisons with other teleosaurids species. This is done in a few places, but not consistently throughout the manuscript. Also, some elements (dorsal vertebrae, ribs, ventral armor) have not any comparison with other taxa.

Discussion:
The assumption of a ventral deflection of the distal tail on M. fitosi is only based on one incomplete and isolated caudal vertebra. I not convinced that there is enough evidence to support this hypothesis. In my opinion, the authors need to make a more persuasive case about that.

---

## Round 0.2 · Minor Revisions

A number of citations in the bibliography are not correctly formatted - one has the title marked in quotation marks, a number have the title partially capitalized, and there are spelling errors as well. As there is no copy-editing the authors must fix these issues.

One anatomical point: There is no such thing as "a left mandible." All tetrapods have a single mandible composed of two rami or hemimandibles. Please use the correct terminology.

---

## Round 0.3 · accepted · Accept

The revised version of the manuscript will now be recommended for Acceptance for publication.

#